# The Relationship between Physical Activity and Long COVID: A Cross-Sectional Study

**DOI:** 10.3390/ijerph19095093

**Published:** 2022-04-22

**Authors:** Jack Wright, Sarah L. Astill, Manoj Sivan

**Affiliations:** 1School of Biomedical Sciences, Faculty of Biological Sciences, University of Leeds, Leeds LS2 9JT, UK; s.l.astill@leeds.ac.uk; 2Academic Department of Rehabilitation Medicine, Leeds Institute of Rheumatic and Musculoskeletal Medicine, University of Leeds, Leeds LS2 9JT, UK; m.sivan@leeds.ac.uk; 3Leeds Teaching Hospitals NHS Trust, Leeds LS9 7TF, UK; 4Community Healthcare NHS Trust, Leeds LS12 5SG, UK

**Keywords:** post-COVID syndrome, COVID-19, post-exertional malaise, pacing, rehabilitation, exercise

## Abstract

The relationship between Long Covid (LC) symptoms and physical activity (PA) levels are unclear. In this cross-sectional study, we examined this association, and the advice that individuals with LC received on PA. Adults with LC were recruited via social media. The New Zealand physical activity questionnaire short form (NZPAQ-SF) was adapted to capture current and pre-COVID-19 PA levels and activities of daily living (ADLs). Participants reported how PA affected their symptoms, and what PA recommendations they had received from healthcare professionals and other resources; 477 participants completed the survey. Mean age (SD) was 45.69 (10.02) years, 89.1% female, 92.7% white, and median LC duration was 383.5 days (IQR: 168.25,427). Participants were less active than pre-COVID-19 (26.88 ± 74.85 vs. 361.68 ± 396.29 min per week, *p* < 0.001) and required more assistance with ADLs in a 7-day period compared to pre-COVID-19 (2.23 ± 2.83 vs. 0.11 ± 0.74 days requiring assistance, *p* < 0.001). No differences were found between the number of days of assistance required with ADLs, or the amount of PA, and the different durations of LC illness (*p* > 0.05). Participants reported the effect of PA on LC symptoms as: worsened (74.84%), improved (0.84%), mixed effect (20.96%), or no effect (28.72%). Participants received contradictory advice on whether to be physically active in LC. LC is associated with a reduction in PA and a loss of independence, with most participants reporting PA worsened LC symptoms. PA level reduction is independent of duration of LC. Research is needed to understand how to safely return to PA without worsening LC symptoms.

## 1. Introduction

Long COVID (LC) can be defined as new or enduring symptoms >4 weeks after an acute COVID-19 infection that cannot be explained by another cause [1,2]. Common LC symptoms include fatigue, shortness of breath, chest pain, musculoskeletal pain and cognitive dysfunction [3]. LC is estimated to have affected 2 million individuals in the UK [4], with at least 10% of non-hospitalised individuals reporting one or more LC symptom at 12 weeks, and a higher prevalence observed in those admitted to hospital with COVID-19 [5]. However, LC is more than just persistent symptoms, it is associated with a high degree of disability. For example, Ziauddeen et al. [3] noted 32% of 2550 participants surveyed required assistance with activities of daily living (ADLs) (activities required for independent living such as eating, dressing and toileting) 6 weeks post-COVID-19 infection [6]. This loss of independence could result in a reduced quality of life [7] and mental health problems [8]. LC also affects people’s ability to return to work, with Davis et al. [1] noting that 45.2% of 3762 participants were working at a reduced capacity and 22.3% were not working due to their health state associated with LC. Taken together, this highlights how LC is likely to have enduring, negative consequences for both social care and the economy, and suggests the need for research on LC management and recovery to prevent this negative sequela.

While the optimal management of LC has not yet been identified, physical activity (PA) may have a role [9]. PA has an array of health benefits including improved sleep, mood and chronic pain; all of which are worsened by LC [1,9,10]. Furthermore, given immune dysfunction is observed in individuals with LC [11], PA’s ability to enhance immune competency provides a physiological mechanism to which PA may facilitate LC recovery [12]. However, current observational studies paint a less conclusive picture. A qualitative study by Humphreys et al. [13] described the positive outcomes on participants mental health when they completed ADLs and outdoor activities. In contrast, Davis et al. [1] reported that 70.7% of participants noted a worsening of LC symptoms and/or a relapse because of PA, but did not explore the type or intensity of PA or its effect on individual symptoms. It could be that inconsistent advice given by healthcare professionals (HCPs) [13] or the lack of guidance on how to resume PA safely [14] resulted in individuals with LC undertaking PA at an inappropriate intensity, which exacerbated symptoms. The research to date suggests there is a complex interaction between PA and LC, including how PA may have the potential to both improve and worsen LC symptoms. However, the relationship between type and intensity of PA and its effect on LC symptoms remains underexplored. Furthermore, there has been little consideration of the effect of LC duration on an individuals’ ability to undertake ADLs despite reports that many with LC require assistance with ADLs [3].

Given the overlap in symptomology, parallels have been drawn between LC and Chronic Fatigue Syndrome (CFS) [15]. While PA is recommended in the management of CFS [16], the type of PA recommended has been subject to controversy [15]. A comparative study of 18,093 participants with CFS across 11 surveys noted that pacing (limiting physical and metal activities to stay within your energy reserves with the aim of avoiding symptom exacerbation) improved symptoms in 44–82% of participants, whereas graded exercise therapy (GET) (incremental increases in PA with the aim of improving exercise tolerance) worsened symptoms in 54–74% [17]. In the UK, the standard recommendation to those recovering from COVID-19 is pacing [18,19]. It could therefore be expected that individuals with LC have reduced PA levels; however, this reduction in PA is yet to be quantified.

Given the above, the aims of this study were (1) to investigate PA patterns in people with LC of varying durations and its relationship to LC symptoms and (2) to capture the type of PA recommended to and undertaken in individuals with LC. It is hypothesised that (1) PA levels will have decreased in both duration and intensity, and the assistance required with ADL will have increased in individuals with LC compared to their pre-COVID-19 baselines; (2) That those who have had LC for the longest will be more physically active and require less assistance with ADL than those who have been more recently diagnosed; (3) LC symptomology will be affected by PA, with the direction of change related to specific PA and symptom factors; and (4) The advice received and PA strategies employed by individuals will be inconsistent. 

## 2. Materials and Methods

### 2.1. Study Design and Participants

In this cross-sectional study, participants (18 yrs+) were recruited through social media using convenience and snowball sampling methods. Given mass testing was not available at the beginning of the pandemic we included those with LC symptoms after suspected COVID-19 irrespective of SARS-CoV-2 diagnostic test results. This is consistent with the current National Institute for Health and Care Excellence (NICE) guidelines that do not require a positive diagnostic test for diagnosis [2]. Participants must have had ongoing symptoms consistent with LC at the time of the completion of the questionnaire and be at least 28 days since COVID-19 infection. Exclusion criteria were lacking the capacity to consent, full recovery of symptoms, and inability to complete the survey in English.

### 2.2. Procedures

An online questionnaire (Onlinesurveys) was designed by the research team, then reviewed by three individuals with LC and refined based on their feedback. Specifically, the volume of text was reduced, the questions streamlined, and the number of free-text answers minimised to prevent fatigue-related dropout. Participants completed the questionnaire between 21 April and 21 May 2021. During this time, the UK was under various restrictions imposed by COVID-19; however, there was unrestricted use of outdoor activity and leisure facilities such as gyms (except in Scotland where gyms reopened on 26th April). Participants provided demographic information, pre-COVID-19 health status, the date they first had COVID-19 symptoms, if and how their COVID-19 infection was confirmed and all LC symptoms they had experienced in the last 7 days. The New Zealand Physical Activity Questionnaire Short Form (NZPAQ-SF) was used to capture participants’ PA patterns (frequency, duration, and intensity) and is a validated and reliable tool for measuring PA in the adult population [20,21]. Furthermore, the NZPAQ-SF is shorter compared to the International Physical Activity Questionnaire (IPAQ), thus the authors felt that this would improve recall and response rates in individuals with LC. The NZPAQ-SF was adapted to explore how much support participants needed when completing ADLs and to compare their PA levels in the last 7-day period at the time of the survey and pre-COVID-19 illness. Participants listed the symptoms that were improved and/or worsened for each of the four PA intensities (ADLs, brisk walking, moderate physical activity (MPA) and vigorous physical activity (VPA)) that was undertaken in the last 7 days. Finally, participants reported the recommendation they received from healthcare professionals (HCPs, such as doctors, nurses and physiotherapists) on PA, including the origin of that information.

### 2.3. Dependent Measures and Statistical Analyses

LC duration was calculated from when participants first reported COVID-19 symptoms (including the first 4 weeks of their illness) and grouped at 6-month intervals. Participants’ PA levels (minutes per week) were calculated by multiplying the number of days per week they were active by the average number of minutes they were active per session for each PA intensity (brisk walking, MPA, VPA). To provide more context to the changes in PA levels, the participant’s PA levels pre- and post-COVID-19 were compared to the UK PA guidelines i.e., completed over 150 min of MPA, 75 min of VPA, or any combination of the two in a week [22]. A McNemar’s test was performed to explore if there was a difference in the number of participants meeting the PA guidelines in the last 7 days compared to their pre-COVID-19 baseline.

A repeated measures (RM) ANOVA explored if assistance required with ADLs had changed because of LC. The model had a within-subject factor of time (the number of days they required assistance at the time of the survey vs pre-COVID-19, denoted as pre-post) and a between-subject factor of LC duration (0–6, 6–12 and 12–18 months). Participants’ responses on questions regarding ADLs were also dichotomised into independent (0 days of assistance per week) and dependent (≥1 day(s) of assistance per week) and a McNemar’s test was performed to explore differences in dependence in the last 7 days compared to participants pre-COVID-19 baseline. To explore if PA patterns changed because of LC an RM ANOVA was performed. The model had within-subject factors of time (PA levels pre vs post-COVID-19, denoted as pre-post) and intensity (brisk walking, MPA, VPA) and a between-subject factor of LC duration. For all RM ANOVAs effect sizes (partial eta) were calculated. Appropriate post hoc inferential testing was performed on significant results from the RM ANOVAs.

Continuous data was described by the mean and standard deviation (SD) if normally distributed and the median and interquartile range (IQR) if non-normally distributed, as determined by the Shapiro-Wilk test. Categorical data was described by percentages and frequencies. Missing data was removed in a pairwise fashion. All statistical analyses were undertaken with SPSS (IBM, Leeds, UK) 26 and an alpha level of *p* ≤ 0.05 was accepted for all inferential statistics.

## 3. Results

Four hundred and ninety-six participants completed the survey. However, 19 participants were excluded as they did not meet the inclusion criteria (16 were <18 years of age, one had symptoms <28 days, one did not complete the survey in English and one had no symptoms in the preceding 7 days). Thus, data from 477 participants was analysed. Participants were predominantly white (92.7%), female (89.1%), and with a mean age of 45.69 (±10.02 years). The median duration of symptoms was 383.5 days (IQR:168.25,427), 51.78% of participants had a COVID-19 infection 12–18 months ago, and 48.2% of participants reported no previous medical conditions (see Table 1). LC symptoms experienced by participants in the last 7 days, including their frequency, are shown in Appendix A.

A summary of the RM ANOVAs used to explore how LC changed participants’ PA levels are available in Table 2, with significant main effects and interactions explored further using post hoc analyses.

Analyses showed participants required significantly more help to complete ADLs in the last 7 days compared to a typical week pre-COVID-19 (2.23 ± 2.83 days vs. 0.11 ± 0.74 days) (t(475) = 16.44, *p* < 0.001), with significantly more people classed as dependent (requiring assistance on ≥1 day per week) in the last 7 days (48.53%) compared to pre-COVID-19 (2.94%) (t(1) = 213.04, *p* < 0.001), as shown in Figure 1.

Participants were less active in the last 7 days (26.88 ± 74.85 min per week) compared to a typical week pre-COVID-19 (361.68 ± 396.29 min per week) (t(476) = 18.75, *p* < 0.001). Post hoc analyses of the main effect of intensity showed that people undertook less VPA (95.23 ± 159.34 min per week) than brisk walking (221.45 ± 257.8 min per week) or MPA (266.15 ± 395.83 min per week, *p* < 0.001), and less brisk walking than MPA (*p* = 0.05).

To explore the interaction between time × intensity, one-way ANOVAs were first used. These showed that pre-COVID-19, participants completed significantly less VPA (186.36 ± 316.36 min per week) than brisk walking (418.56 ± 507.18 min per week) or MPA (480.11 ± 715.06 min per week, *p* < 0.001), and equal amounts of brisk walking and MPA (*p* = 0.23). In the last 7 days, participants did significantly more MPA (52.19 ± 195.17 min per week) than brisk walking (24.33 ± 62.76 min per week) or VPA (4.11 ± 22.95 min per week) (*p* = 0.001 and *p* < 0.001 respectively), and more brisk walking than VPA (*p* = 0.03). Paired t-tests showed that, on average, participants completed significantly less of each PA intensity in the last 7 days than their pre-COVID-19 baseline, as shown in Table 3.

Analyses showed that significantly more participants met the UK PA guidelines pre-COVID-19 (83.65%) compared to in the last 7 days (8.18%) (t(1) = 356.03, *p* < 0.001), with 244 participants (51.12%) reporting they had not done any MPA or VPA in the last 7 days. Excluding these participants showed that 16.74% of the remaining 233 participants met the UK PA guidelines.

PA from one or more PA intensities (ADLs, brisk walking, MPA or VPA) worsened LC symptoms in 74.84% of participants (357/477), improved them in 0.84% (4/477), had a mixed effect (the ability to both improve and worsen symptoms) in 20.96% (100/477) and had no effect in 28.72% (137/477) (see Figure 2). Data was missing for ADLs in 9 participants. For brisk walking, MPA and VPA, participants could only respond if they had undertaken that PA of that intensity in the last 7 days, and therefore only 163 responses were available for brisk walking, 218 for MPA and 33 for VPA. Participant characteristics were explored for each effect group (worsened, improved, mixed, no effect) (see Appendix A. In the worsened group there was a higher proportion of women, a lower proportion met the UK PA guidelines post-COVID-19, and on average they had more LC symptoms in the last 7 days, compared to the other groups.

Across all PA intensities, the most worsened symptoms by engaging in PA were fatigue (68.4%), respiratory symptoms (56.23%), and musculoskeletal symptoms (42.72%), Figure 3 shows 11 symptoms that were most frequently worsened by PA. The remaining symptoms that were worsened by PA are shown in Appendix A. Of those that found an improvement, the most improved symptoms were mental health (36.06%), musculoskeletal symptoms (19.38%), and fatigue (15.61%) (see Appendix A)

Healthcare professionals (HCPs) encouraged PA in 45.7% of participants. In contrast, 28.3% of participants reported that HCPs discouraged PA. Other resources that participants used to guide their decision provided inconsistent advice (see Figure 4).

HCPs recommended 16 distinct PA strategies (see Figure 5). However, when participants were asked what PA strategies they had tried as part of their recovery, 78% had used pacing, with only 22.6% using GET (graded exercise therapy).

## 4. Discussion

The main findings of this study were an association between LC and a reduction in PA levels and an increase in the amount of assistance required for ADLs (see Appendix B). Furthermore, whilst PA tended to make LC symptoms worse, a small number of individuals reported an improvement in symptoms with certain PA intensities. Participants received conflicting information from HCPs and other resources used to inform their decisions on PA.

### 4.1. Physical Activity Patterns and Independence

There was no observed association between the different durations of LC illness and PA and/or independence levels. This is contradictory to the belief that those with a longer duration since their acute illness would have managed to return to their pre-COVID activity levels. However, this finding should be interpreted with caution as our study used a self-selected sample, and individuals struggling to keep up with their PA levels may have been more motivated to participate in this study. A longitudinal study would be a better design to capture such temporal associations during the course of LC.

While low levels of PA were expected in this population, it could reflect attempts to avoid Post Exertional Malaise (PEM) (the worsening of symptoms after physical, mental, or emotional exertion) [23]. In addition to low levels of PA, participants may spend more time engaging in sedentary behaviour (SB) (any waking behaviour that requires an energy expenditure ≤1.5 metabolic equivalents) [24]. Both physical inactivity (not meeting PA guidelines) and SB are independent risk factors for a multitude of ill health effects including cardiovascular disease, depression, reduced quality of life, and all-cause mortality [25,26,27]. Given our data shows that PA levels do not seem to increase with the duration of LC (see Table 2), it could be that individuals with LC are at an extended risk of the negative consequences of physical inactivity and SB and highlights the need to promote the safe return to PA in this population.

Facilitating a return to independence should be a priority of HCPs and rehabilitation programs to prevent the secondary consequences of a loss of independence such as a reduced quality of life [7] and depression [8]. It is unknown if the additional assistance required by those with LC is provided formally (healthcare services) or informally (friends or family). However, as the incidence of COVID-19 continues to rise [28], so too will the incidence of LC, and this will cause a significant demand on social care, families, and friends, and negative consequences for local and national economies [29].

### 4.2. PA’s Effect on LC

While we explored the characteristics between effect groups (worsened, improved, mixed, no effect) (see Appendix A), comparisons were limited by the small sample sizes of many of the groups. Mental health was the symptom most frequently improved by PA and was reported by 7.55% (36/477) participants which is consistent with the findings of Humphreys et al. [13]. Given a loss of independence and physical inactivity, both consequences of LC, are associated with depression, finding out why these participants benefit from PA and translating it to the wider LC population could help prevent a mental health crisis in this population.

Data showed that 20.96% of participants reported a mixed effect of PA on their symptoms. It is unclear if this represents a simultaneous worsening and improvement of different symptoms or if the same symptom gets better and worse at different points in time. Given some symptoms are both improved and worsened by PA (e.g., fatigue) this may support the latter explanation. Alternatively, currently unknown intrapersonal factors such as the severity of COVID-19 illness or pre-COVID-19 fitness levels could cause the same symptom to worsen in some but improve in others.

### 4.3. Recommendations by HCPs

While pacing is recommended to individuals with LC [18,19], it was only the 4th most recommended strategy by HCPs (see Figure 5). In contrast, GET was the 2nd most recommended strategy. The NICE guidelines on CFS have recently been updated and no longer recommended GET [16] due to its potential to worsen symptoms [17]. Given GET may also worsen LC symptoms [15], participants are potentially receiving harmful recommendations from HCPs about how to be physically active. Seventeen percent of participants report that they were recommended slow or gentle PA without explicitly stating they were recommended pacing. This may be as HCPs and/or participants do not clearly understand the definition of pacing. Alternatively, this may represent a miscommunication between HCPs and their patients. Looking forward, more effort needs to be made in supporting HCPs to recommend PA in line with new recommendations on how individuals with LC engage in and increase PA levels (see Appendix B) [30]. More specifically, people with LC that do not experience PEM should gradually increase their level of activity through five phases of increasing intensity [30].

### 4.4. Limitations

While the use of social media allowed data to be captured from a large sample during a time of COVID-19 restrictions, it does have several limitations. Firstly, it excludes those who are digitally illiterate or do not use social media. Furthermore, this was a self-selected sample rather than one that is more representative of the wider LC population. As such, our findings may not be applicable to those not engaging with social media or from ethnic minorities. However, our sample was comparable to other research with regards to pre-existing health conditions, symptom burden and the pattern of LC symptoms [1,31]. In addition, while the NZPAQ-SF is a validated tool for measuring PA in the adult population, it has not been validated to determine the amount of assistance required with ADLs or collecting data from more than 7 days ago. As 78% of participants were using pacing, their activity may not have met the rigid criteria of the NZPAQ-SF. For example, only brisk walking is included in the NZPAQ-SF, whereas an individual using pacing may have purposefully walked at a slow pace to conserve their energy and prevent a worsening of their symptoms/PEM. Therefore, the NZPAQ-SF may have underestimated participants’ total PA by excluding light physical activities [21].

### 4.5. Future Directions

Research is needed to explore the complex interaction between PA and LC symptomology, including why some individuals benefit from PA and others do not. The development of individualised PA programmes that could mitigate the negative health consequences of physical inactivity without worsening LC symptoms and facilitate a return to independence should be considered a clinical priority. This research also highlights the need for policymakers to: limit the transmission of COVID-19, which consequently will reduce the incidence of LC and mitigate its negative effects on healthcare and the economy; prepare and plan for an increased demand on social care; and ensure HCPs are provided with guidance to facilitate the clear, safe care of their patients.

## 5. Conclusions

To the best of our knowledge, this is the first study to quantify the changes in PA and ADLs because of LC and explore the dynamic interaction between PA and LC symptomology. LC causes drastic reductions in the ability to be physically active and to complete ADLs independently, which could be explained by the observation that LC symptoms worsen with increased PA. In a minority of people, PA can improve symptoms or has a mixed effect. This suggests a cautious approach as increasing PA may be warranted, but how to do this, and for who and when, needs to be further explored.

## Figures and Tables

**Figure 1 ijerph-19-05093-f001:**
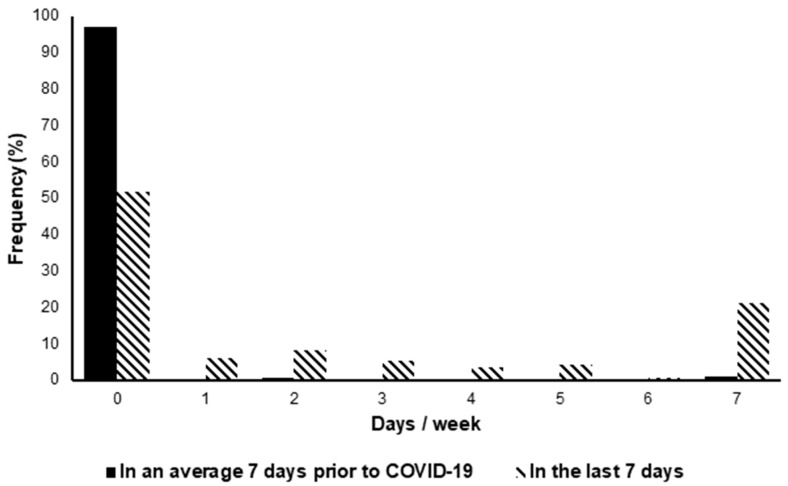
The number of days participants required assistance with ADL (activities of daily living) in the last 7 days compared to their pre-COVID-19 baselines.

**Figure 2 ijerph-19-05093-f002:**
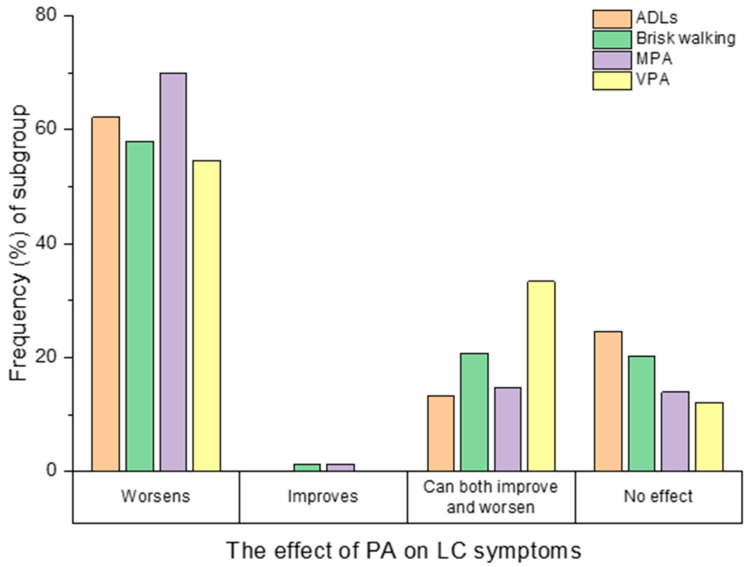
How completing activities of daily living (ADLs, *n* = 468), brisk walking (*n* = 169), moderate physical activity (MPA, *n* = 223) and vigorous physical activity (VPA, *n* = 33) effected participants Long COVID (LC) symptoms.

**Figure 3 ijerph-19-05093-f003:**
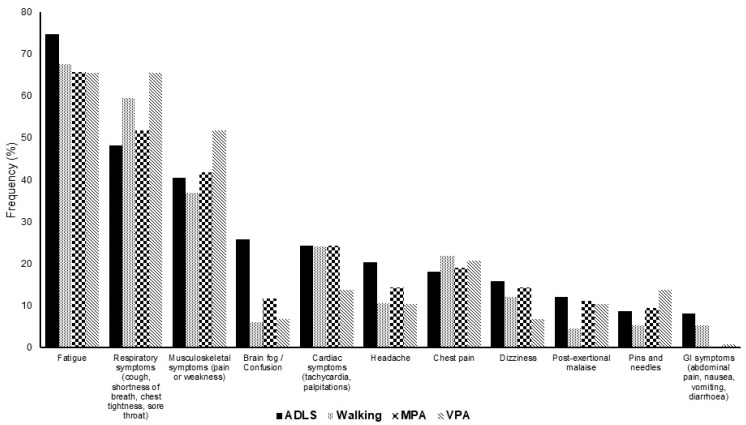
The frequency (%) of the 11 LC symptoms most commonly worsened by activities of daily living (ADL, *n* = 353), brisk walking (*n* = 133), moderate physical activity (MPA, *n* = 189) and vigorous physical activity (VPA, *n* = 29).

**Figure 4 ijerph-19-05093-f004:**
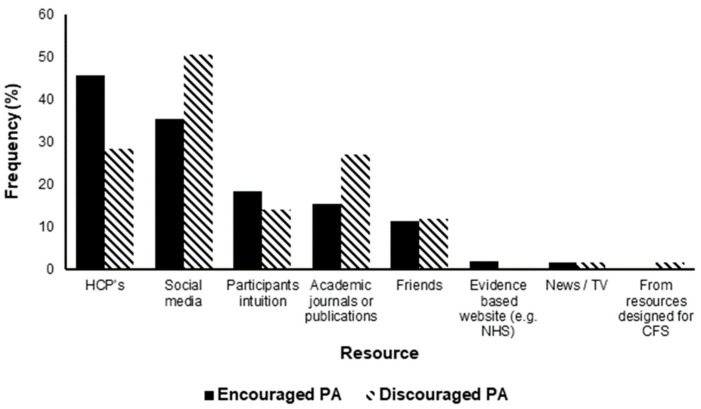
The frequency (%) that resources have encouraged and/or discouraged physical activity (PA) in 477 participants with Long COVID (LC). CFS; chronic fatigue syndrome, HCPs; healthcare professionals.

**Figure 5 ijerph-19-05093-f005:**
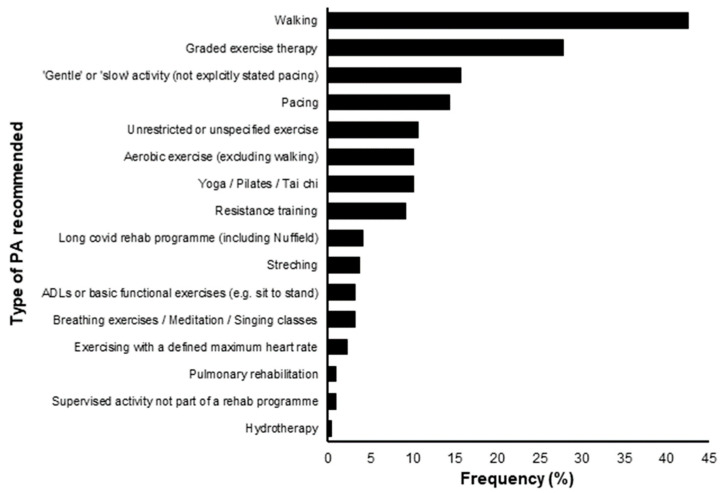
The frequency (%) of which different types of physical activity (PA) were recommended to participants by healthcare professionals in a sample of 216.

**Table 1 ijerph-19-05093-t001:** Participant characteristics.

Characteristic	Total Sample (*n* = 477)
Age (years), mean (SD)	45.69 (10.02)
Gender (female), *n* (%)	425 (89.10)
BMI, median (IQR)	25.71 (22.51, 30.47)
Ethnicity, *n* (%)	
White (British, Irish, Irish Traveller or other White backgrounds)	442 (92.70)
Black (African, Caribbean or other Black backgrounds)	3 (0.60)
Asian (Indian, Pakistani, Bangladeshi, Chinese or other Asian backgrounds)	18 (3.80)
Mixed (White and Asian, White and Black African, White and Black Caribbean, Other)	7 (1.50)
Other	4 (0.80)
Country, *n* (%)	
England	358 (78.34)
Scotland	43 (9.41)
Wales	26 (5.69)
USA	11 (2.41)
Canada	5 (1.09)
Northern Ireland	5 (1.09)
Ireland	3 (0.66)
Finland	2 (0.44)
France	1 (0.22)
India	1 (0.22)
Netherlands	1 (0.22)
Sweden	1 (0.22)
Number of LC symptoms, median (IQR)	11 (8,14)
Time since COVID-19 symptom onset (months), *n* (%)	
0–6	132 (27.67)
6–12	91 (19.08)
12–18	247 (51.78)
Method of COVID-19 diagnosis, *n* (%)	
PCR test	226 (47.4)
Antibody test	50 (10.5)
Based on symptoms alone (including retrospectively)	177 (37.1)
No testing available at the time	12 (2.5)
Other	7 (1.5)
Co-morbidities prior to LC, *n* (%)	
Allergies *	12 (2.5)
Autoimmune diseases	42 (8.8)
Cardiovascular disease	20 (4.2)
Chronic neurological conditions	10 (2.1)
Chronic pain	13 (2.7)
Chronic respiratory conditions	94 (19.7)
Diabetes (type 1 or 2)	17 (3.6)
Mental health **	12 (2.5)
Migraines	10 (2.1)
No diagnosed co-morbidities	230 (48.2)
Osteoarthritis	11 (2.3)
Other (any co-morbidity with a frequency of <2%)	64 (13.4)
Unspecified hypo or hyperthyroidism	12 (2.5)

*n*; number, PCR; polymerase chain reaction, SD; standard deviation. * Allergies includes hay fever, eczema, coeliac disease, and non-coeliac gluten sensitivity. ** Mental health includes anxiety, depression, PTSD, and bipolar affective disorder.

**Table 2 ijerph-19-05093-t002:** A summary of the main effects from the repeated measures ANOVA.

	F	*df*	*p*	η2
**ADLs**				
Pre-post	225.97	1467	<0.001	0.32
Pre-post × LC duration	0.16	2467	0.86	0.01
LC duration ^Ϯ^	0.20	1467	0.82	0.00
**PA**				
Pre-post	286.31	1467	<0.001	0.38
Pre-post × LC duration	0.11	2467	0.89	0.00
Intensity	51.67	1.61, 751.29	<0.001	0.10
Intensity × LC duration	1.83	3.22, 751.29	0.14	0.01
Pre-post × intensity	36.85	1.72, 802.36	<0.001	0.07
Pre-post × intensity × LC duration	0.61	3.44, 802.36	0.63	0.00
LC duration ^Ϯ^	0.13	2467	0.88	0.00

ADLs; activities of daily living, df; degrees of freedom, LC; Long COVID, η2; ETA squared, *p*; significance, PA; physical activity. Ϯ between-subject factor.

**Table 3 ijerph-19-05093-t003:** The difference between participants activity in the last 7 days compared to their pre-COVID-19 baselines.

Intensity	Minutes per Week, Mean (SD)		Mean Difference (95% CI)	Paired *t*-Test, *p* Value
	Pre-COVID-19 Baseline	In the Last 7 Days		
Brisk walking	418.56 (507.18)	24.33 (62.76)	394.23 (348.67–439.8)	<0.001
MPA	480.11 (715.06)	52.19 (195.17)	427.92 (366.11–489.74)	<0.001
VPA	186.36 (316.36)	4.11 (22.95)	182.25 (153.84–210.65)	<0.001

CI; confidence interval, MPA; moderate physical activity, *p*; significance, SD; standard deviation, VPA vigorous physical activity.

## Data Availability

Anonymised data sets will be made available at the request of journal reviewers, other researchers, interested parties, and/or potential collaborators on an individual request basis. We have consent from participants to do so. It will be stipulated that any external users must not attempt to identify any study participants or breach confidentiality.

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
