# Peer review of "The Relationship between Physical Activity and Long COVID: A Cross-Sectional Study"

_ijerph, 2022, doi:10.3390/ijerph19095093_

Round 1

Reviewer 1 Report

This exploratory study, highlights the problems related to Long-Covid, physical function and physical activity in a rather young, predominantly female population. I find this a very important topic and worthy of accurate investigation. Despite the limitations of the study design, these data provide important information and emphasize the need for specific guidelines and intervention protocols that cannot be one-size-fits-all kind of recommendations.

Author Response

We thank the reviewer for the overwhelmingly positive responses to our paper. 

Reviewer 2 Report

This cross-sectional study used a survey method to collect data to probe the relationship between Long Covid (LC) symptoms and physical activity (PA) levels. The sample size (N=477) is quite large. The topic is quite interesting because it is related to subjects’ physical activity in the Long Covid 19 pandemic. However, it is not a novelty. I have a few minor suggestions that I hope you will find useful.

 - Abstract: The abstract does not follow the editorial standard indicated for IJERPH: “The abstract should follow the style of structured abstracts, but without headings”; therefore, remove the words: Introduction, Methods, Results and Conclusions.

 - A zero should not be inserted before a decimal fraction when the number cannot be greater than 1. For example, p < 0.05 should be written as “p < .05.” Continues in the same way!

Typically, if the exact p value is less than .001, you can merely state p < .001.

You have to move Strengths and limitations of this study at the end, before conclusions. Delete the section next abstract.

 - In Introduction many parts of the literature review only simply put previous studies together without themes. So, readers couldn't get a clue of focus. What do authors want to tell readers?

 You should follow the style of reference in the text. The style is wrong.

 - The use of online survey could carry the possibility of double compilation. How the authors control this source of bias?

In the section "procedure", you have added “measures and procedure”

The choice of measures is not very clear. Why you didn't use validated scales?

What is the reliability and the Cronbach of NZPAQ-SF???

  - Eligibility of the population is missing, have you performed a regression study? I recommend that you rely on the CROSS guidelines

https://www.researchgate.net/publication/353522424_Checklist_for_Reporting_Of_Survey_Studies_CROSS

 You have to add the reference of the version SPSS, the city.

Author Response

Point 1. The abstract does not follow the editorial standard indicated for IJERPH: “The abstract should follow the style of structured abstracts, but without headings”; therefore, remove the words: Introduction, Methods, Results and Conclusions.

Response 1. Thank you for noting this, we have removed the headings removed in line with editorial standard

Point 2. A zero should not be inserted before a decimal fraction when the number cannot be greater than 1. For example, p < 0.05 should be written as “p < .05.” Continues in the same way!

Response 2. We have made changes throughout the manuscript to reflect above recommendation, thank you

Point 3. You have to move Strengths and limitations of this study at the end, before conclusions. Delete the section next abstract.

Response 3. We have removed the section next to the abstract

Point 4. In Introduction many parts of the literature review only simply put previous studies together without themes. So, readers couldn't get a clue of focus. What do authors want to tell readers?

Response 4. Thank you for the suggestion. We have re-structured and re-written the introduction to try and focus readers to the gaps in the current LC literature and the questions we aim to answer. For example, we now note how previous literature has shown both positive and negative effects of PA on LC. We suggest this discrepancy may be due to the varying intensities of PA which were not explored in previous studies. We also highlight different PA regimes in the management of LC drawing parallels with the CFS literature. We have removed any text that discussed LC but was not directly relevant to our study. To clarify our study aims we have also included our hypotheses.

Point 5.  You should follow the style of reference in the text. The style is wrong.

Response 5. The reference style has been amended to reflect the IJERPH guidelines, thank you

Point 6.  The use of online survey could carry the possibility of double compilation. How the authors control this source of bias?

Response 6. Thank you for your question. We checked for duplication of results when the data was cleaned and none was identified.

Point 7. In the section "procedure", you have added “measures and procedure”. The choice of measures is not very clear. Why you didn't use validated scales?

Response 7. Thank you for your question. The NZPAQ-SF has been validated against heart rate monitors as a tool for measuring PA (https://pubmed.ncbi.nlm.nih.gov/18364525/). It has also been validated against the IPAQ and doubly labelled water (https://www.ncbi.nlm.nih.gov/pmc/articles/PMC2219963/). The NZPAQ-SF is actually a modified version of the IPAQ-SF however, the we decided to use the NZPAQ-SF because it has less questions, did not ask questions irrelevant to our study (the IPAQ-SF asked about sedentary behaviour) and therefore would be easier and quicker for participants to complete. We were particularly mindful of the duration of the survey as many indiviudals with LC have difficulties with concentration and/or had exacerbations of symptoms from mental activities. The importance of a simple and clear survey was re-iterated to use by the 3 individuals with LC who reviewed the survey prior to its launch.

While there are many validated tools for measuring ADLs (https://www.ncbi.nlm.nih.gov/pmc/articles/PMC7320974/), these tools were designed for specific populations such as the older adult, post-stroke, or in those with dementia – none of which were our population of interest. Furthermore, the aim of our study was to explore the amount of assistance individuals required with ADLs as a marker for independence as opposed to directly measuring individual ADLs. More specifically, we aimed to explore how the amount of assistance required with ADLs changed over time (pre vs post COVID-19 and then with increasing durations of LC), for which there is not a validated tool. In the interest of survey duration and complexity we added two questions about assistance with ADLs into the survey which mirrored the structure already present in the NZPAQ-SF.

Point 8. What is the reliability and the Cronbach of NZPAQ-SF???

Response 8.  The test–retest reliability of the NZPAQ-SF ranges from 0.33 to 0.69 (https://pubmed.ncbi.nlm.nih.gov/18053188/  
https://pubmed.ncbi.nlm.nih.gov/18364525/

https://pubmed.ncbi.nlm.nih.g[1]ov/18981036/ )

The reliability of the NZPAQ-SF has been added to the methods section.

Point 9.  Eligibility of the population is missing, have you performed a regression study? I recommend that you rely on the CROSS guidelines

Response 9. Thank you for your suggestion. The eligibility criteria of our participants are defined in the methods. Two participants (0.4%) were excluded as they did not meet the definition of LC (one had symptoms for less than 4 weeks, the other had no symptoms in the preceding 7 days). We have completed a CROSS checklist.

Reviewer 3 Report

Wright, Astill, Sivan: The Relationship Between Physical Activity and Long Covid: A cross-sectional study

Dear Authors,

I like the results of this study on the associations between Long Covid and physical activities. This MS can be published as it stands. I have only one comment. I noticed that the participants of this study had several co-morbidities before LC, such as autoimmune diseases, diabetes, and cardiovascular disease. High body fat levels and obesity are considered physiological traits affecting the energy homeostasis system linking body fat reserves with cardiovascular disease, diabetes, etc.

On the other hand, different types of fat may have mixed effects on COVID-19 infection rates and outcomes (e.g., Krams et al. 2021 Int. J. Environ. Res. Public Health, https://doi.org/10.3390/ijerph18031029). Have you any data on obesity levels? If you have, it would be great to see them. If not, could you mention such associations in the Discussion, please?

Author Response

Point 1. Have you any data on obesity levels? If you have, it would be great to see them. If not, could you mention such associations in the Discussion, please?

Response 1.  Thank you for your review and insight into the relationship between obesity and acute COVID-19 infections. We have BMI data for 449 participants, and this has been included in the participant characteristics table. Interestingly, the median BMI of our sample was lower than that of the average UK adult making it hard to draw conclusions between obesity and LC.

Round 2

Reviewer 2 Report

this revised manuscript is suitable for submission